# K-Ras Binds Calmodulin-Related Centrin1 with Potential Implications for K-Ras Driven Cancer Cell Stemness

**DOI:** 10.3390/cancers15123087

**Published:** 2023-06-07

**Authors:** Ganesh babu Manoharan, Christina Laurini, Sara Bottone, Nesrine Ben Fredj, Daniel Kwaku Abankwa

**Affiliations:** Cancer Cell Biology and Drug Discovery Group, Department of Life Sciences and Medicine, University of Luxembourg, L-4362 Esch-sur-Alzette, Luxembourg

**Keywords:** K-Ras, centrin, calmodulin, mitosis, centrosome, BRET

## Abstract

**Simple Summary:**

Trafficking chaperones facilitate the spatio-temporal distribution pattern of proteins inside cells. In the case of the membrane-anchored protein Ras, trafficking chaperones typically bind to the C-terminal farnesyl-moiety. Thus shielded from the aqueous environment, Ras can diffuse more efficiently through the cytoplasm. The calcium-binding protein calmodulin (CaM) was proposed as a K-Ras trafficking chaperone. However, CaM has many different functions inside the cell. Centrin proteins are highly related to calmodulin, and we find that they also bind to K-Ras. Unexpectedly, this interaction depends on the activation state and the effector binding site of K-Ras, not on the farnesyl-anchor. Overall, CaM and centrin1 appear to enable only a fraction of K-Ras membrane anchorage. Given that CaM inhibitors also affect the K-Ras/centrin1 interaction and the very similar distribution of centrin1 and CaM throughout the cell cycle, the dependence of K-Ras on either protein may be difficult to determine.

**Abstract:**

Recent data suggest that K-Ras4B (hereafter K-Ras) can drive cancer cell stemness via calmodulin (CaM)-dependent, non-canonical Wnt-signalling. Here we examined whether another Ca^2+^-binding protein, the CaM-related centrin1, binds to K-Ras and could mediate some K-Ras functions that were previously ascribed to CaM. While CaM and centrin1 appear to distinguish between peptides that were derived from their classical targets, they both bind to K-Ras in cells. Cellular BRET- and immunoprecipitation data suggest that CaM engages more with K-Ras than centrin1 and that the interaction with the C-terminal membrane anchor of K-Ras is sufficient for this. Surprisingly, binding of neither K-Ras nor its membrane anchor alone to CaM or centrin1 is sensitive to inhibition of prenylation. In support of an involvement of the G-domain of K-Ras in cellular complexes with these Ca^2+^-binding proteins, we find that oncogenic K-RasG12V displays increased engagement with both CaM and centrin1. This is abrogated by addition of the D38A effector-site mutation, suggesting that K-RasG12V is held together with CaM or centrin1 in complexes with effectors. When treated with CaM inhibitors, the BRET-interaction of K-RasG12V with centrin1 was also disrupted in the low micromolar range, comparable to that with CaM. While CaM predominates in regulating functional membrane anchorage of K-Ras, it has a very similar co-distribution with centrin1 on mitotic organelles. Given these results, a significant overlap of the CaM- and centrin1-dependent functions of K-Ras is suggested.

## 1. Introduction

*KRAS* is the most frequently mutated oncogene and in addition mutated in congenital disorders, called RASopathies [1,2]. It is not fully understood why *KRAS* is more frequently mutated in cancer than the other *RAS* genes, *NRAS* and *HRAS*. Several facets of Ras biology may contribute to the higher exploitation of *KRAS*, such as its higher expression level, its specific intracellular trafficking and distribution, or its distinct nanoscale organization in the plasma membrane that imposes differential effector usage [3,4,5]. Another less characterized difference is the ability of Ras proteins to drive stemness properties in cells [6,7]. Notably, the most common *KRAS* splice variant, K-Ras4B (hereafter K-Ras), but not H-Ras, mediates stemness properties via calmodulin (CaM)-dependent non-canonical Wnt-signalling [6]. In line with this, CaM inhibitors block the stemness properties of *K-RAS*-mutant cancer cells [8,9]. However, the exact mechanism of how CaM mediates K-Ras-driven stemness is not resolved.

Previous cellular data showed that K-Ras/CaM complexes are disrupted by phosphomimetic mutations of Ser181 at the C-terminus of K-Ras. Conversely, CaM binding blocked phosphorylation at that site [10]. Intriguingly, the phosphomimetic mutation of K-RasG12V on Ser181 reduces its ability to drive stemness [6]. Mutations at this site also modulate the interaction with another trafficking chaperone PDE6D [11], which traffics several prenylated proteins to stemness mediating organelles [12]. Hence, CaM may not be alone in mediating the K-Ras-stemness activity.

CaM possesses two Ca^2+^-binding lobes, which can each encase 15–20 residue long peptide stretches of classical target proteins in their hydrophobic surfaces [13]. Classical target peptides are typically helical, positively charged, and contain hydrophobic anchor residues. Very similar biochemical characteristics are found in singly lipidated, polybasic termini of prenylated or myristoylated proteins, which have emerged as non-canonical targets of CaM [14]. CaM facilitates the Ca^2+^-dependent cytoplasmic solubilization of K-Ras by sequestering its farnesyl-tail from the aqueous environment [15]. This contrasts to the GTP-Arl2/3 triggered release of PDE6D cargo [16]. PDE6D and CaM share the preference for K-Ras amongst the Ras isoforms as palmitoylation obstructs access to the hydrophobic pockets, making K-Ras4A, N-Ras, and H-Ras clients only in their non-palmitoylated states [17,18]. Both trafficking chaperones are found in the cyto- and nucleoplasm and on centriolar structures, such as the primary cilium and the centrosomes [16,19,20]. Hence, it is plausible to assume that these two chaperones have overlapping, yet distinct roles in coordinating trafficking of prenylated proteins spatio-temporally.

In cell lysates, CaM engages more with GTP-loaded K-Ras than with its inactive counterpart [17,21]. Furthermore, complexes between K-Ras, CaM, and PI3K p110 subunits have been proposed as being relevant for Akt activation during platelet-derived growth factor receptor (PDGFR)-mediated cell migration [22,23]. The fact that the activation state of Ras matters for its interaction with CaM contrasts with in vitro and structural data. Only weak transient contacts of CaM with non-farnesylated K-Ras were observed in NMR-experiments, while the farnesylated poly-lysine stretch of K-Ras comprising residues 180–185 was sufficient for CaM binding [15,24]. In vitro data further suggest that the polybasic and farnesylated C-terminus of K-Ras binds to either of the Ca^2+^-bound lobes of CaM, but without involvement of the G-domain [25]. Thus, it appears that the farnesylated C-terminus of K-Ras is sufficient for micromolar binding to CaM. However, in cells, there may be CaM/ K-Ras complexes that depend on the activation state of K-Ras.

Inhibitors of CaM alter its conformation, thus preventing binding of canonical target peptides and non-canonical targets [9,13,26,27]. The covalent CaM inhibitor ophiobolin A disrupts binding of K-Ras to CaM and K-Ras membrane anchorage by irreversibly modifying Lys75, 77, and 148 of CaM [8,9,28]. We recently developed an alternative, less toxic covalent inhibitor of CaM, called Calmirasone1, which is much more suitable for cell biological applications [9].

Centrin (or caltractin) proteins are highly related to CaM with the same bi-lobal structure, however, only the C-terminal lobe binds and senses Ca^2+^ with high affinity [29]. This leaves the centrin-specific N-terminus free for mediating self-assembled extended structures of centrins, which are Ca^2+^-dependent due to allosteric coupling with the C-terminus [30]. In humans, three centrin paralogs (centrin1-3, *CETN1-3*) are known [31]. While centrin2 and centrin3 are ubiquitously expressed, centrin1 expression is limited to male germ cells, neurons, and ciliated cells [32]. Centrin2 is probably best known for binding and stabilizing XPC (xeroderma pigmentosum group C), which is involved in DNA repair [33]. In addition, centrins have been implicated in nuclear pore functions and proteasomal activities [32]. Like CaM, centrins appear to recognize a hydrophobic motif of 15–20 residues in such classical target proteins [34].

The activity of centrins can be regulated by several phosphorylation and SUMOylation events [34]. Nuclear localization of centrin2 is enhanced by its SUMOylation [35]. Phosphorylation of T118 in the third EF-hand of the centrin2 C-terminal lobe is required for Ca^2+^-binding and its centrosomal localisation [32]. Centrins localise to distal and intermediate regions preferentially from the mother centrioles and are part of a set of 14 ancient and highly conserved centriolar proteins [36,37]. Hence, loss of centrins broadly affects centriolar functions, including organisation of the microtubule network or overall biogenesis of centrioles [32]. Based on the essential roles of centrins in uni-cellular organisms that depend on cilia formation, it is plausible to assume that an important role also exists for centrins in vertebrate/mammalian ciliogenesis [32]. In line with this, ciliogenesis is reduced upon depletion of centrin2 in hTERT-RPE1 cells [38].

Given the highly similar bi-lobal structure with hydrophobic binding pockets, we hypothesized that centrins also bind to non-classical targets of CaM, such as K-Ras. Here we show that K-Ras binds to centrin1 in cells in a similar manner to CaM. Our results suggest that binding of K-Ras to these Ca^2+^-binding proteins in cells is largely independent of the prenylation of K-Ras and involves the G-domain. Given that CaM inhibitors also affect the K-Ras/centrin1 interaction and the very similar distribution of centrin1 and CaM throughout the cell cycle, the dependence of K-Ras on either protein may be difficult to determine.

## 2. Experimental Procedures

### 2.1. Plasmids, siRNAs and Inhibitors

All construct names contain the tag at a position corresponding to its location in the protein sequence, e.g., GFP2-CaM, contains the GFP2-tag at the N-terminus of CaM. All plasmids employed in the study were produced by multi-site gateway cloning [39]. The human CaM (*CALM1)* entry clone with L1–L2 recombination sites was obtained from the NCI RAS Initiative. The K-Ras4b entry clone was from RAS mutant clone collection (Kit #1000000089) procured from Addgene (Watertown, MA, USA). Custom-synthesised entry clones encoding human centrin1 (*CETN1)* or the CTK fragment with L1–L2 recombination sites in pDONR221 vector were commercially obtained from Genecust, Boynes, France. An LR recombination reaction comprising three entry clones encoding the CMV promoter, a tag (Rluc8 or GFP2) and the protein of interest (CTK, K-Ras wt, CaM and centrin1); a destination vector, pDest-305 vector, was performed to obtain the recombinant plasmids. In a single-site LR recombination reaction, CaM or centrin1 entry clones were combined with the destination vector, pDest-527, to produce bacterial expression plasmids encoding N-terminally His6-tagged CaM and centrin1. The positive clones were selected using ampicillin in *E. coli* DH10B. The pmCherry-CaM, pEGFP-centrin1 plasmids and plasmids encoding N-terminal Rluc8 or GFP2-tagged K-RasG12V and H-RasG12V were previously described [9,26]. siRNA for *CALM1* (Hs_CALM1_6, SI02224222), and *FNTA* (Hs_FNTA_6, SI02661995) were obtained from Qiagen (Venlo, The Netherlands). The siRNA for *CETN1* (ON-TARGETplus SMARTpool siRNA, L-011831-00-0005) and negative control siRNA (ON-TARGETplus Non-targeting pool, D-001810-10-05) were obtained from Dharmacon (Cambridge, UK). Mevastatin (J61357, Alfa Aesar, Leuven, Belgium), calmidazolium chloride (sc-201494, Santa Cruz, Heidelberg, Germany), and ophiobolin A (sc-202266, Santa Cruz, Heidelberg, Germany) were commercially acquired from the sources given in parenthesis. Calmirasone1 was synthesized as previously described by us [9].

### 2.2. Protein Sequence Analyses

The protein sequences encoded by *CALM1-3* and *CETN1-3* genes were collected from the uniprot database (http://unirprot.org/; last accessed 2 April 2023) and a multiple sequence alignment was performed using Clustal Omega (https://www.ebi.ac.uk/Tools/msa/clustalo/; last accessed 2 April 2023). For paralog number analysis, the protein coding genes of calmodulin and centrin were searched for each species in the NCBI protein database (https://www.ncbi.nlm.nih.gov/; last accessed 2 April 2023). The *CALM1* or *CETN1* genes were given as search query and orthologs were identified from the annotation pipeline. A process flow was then generated using RefSeq to identify a set of comparable proteins including orthologs and similar proteins. Note that only protein encoding genes were considered and pseudogenes were discarded.

### 2.3. Protein Purification

The His6-tagged human CaM and centrin1 proteins were purified as described previously [9]. Briefly, the pDest527-His6-CaM or pDest527-His6-centrin1 plasmid transformed *E. coli* BL21 (DE3) cells were grown in LB medium supplemented with 100 μg/mL of ampicillin. At 0.4–0.6 OD, 0.5 mM IPTG was used to induce the culture with subsequent overnight incubation at 25 °C with shaking. After centrifugation of the culture, its pellet was suspended in a lysis buffer composed of 20 mM HEPES, pH 7.6, 150 mM NaCl, 5 mM MgCl_2_, 0.5 mg/mL lysozyme, and 700 units DNase I. For the pellet from 1 l of cell culture 20 mL of lysis buffer was used. After cell lysis by sonication, the His-tagged proteins were purified using HisTrapTM HP Prepacked Columns (GE Healthcare, Leuven, Belgium) on the ÄKTAprime plus chromatography system (GE Healthcare). A buffer composed of 50 mM Tris HCl, pH 7.5, 150 mM NaCl, and 35 mM imidazole was used to equilibrate the column, and His-tagged proteins were eluted using 250 mM imidazole elution buffer. Afterwards, the eluted fractions were dialyzed for 16 h at 4 °C in dialysis buffer (50 mM Tris HCl, pH 7.5, 150 mM NaCl, and 2 mM CaCl_2_). Using a NanoDrop 2000c Spectrophotometer (Thermo Fisher Scientific, Merelbeke, Belgium), the protein concentration was determined by absorbance. 

### 2.4. Fluorescence Polarisation Binding Assay

Fluorescence polarisation assays were performed as established previously by us [9,26]. The fluorescein-labelled PMCA- and CaMKII-peptides were custom synthesized by Genscript (Piscataway, NJ, USA) and Pepmic (Suzhou, China), respectively. The PMCA peptide was derived from 1086-LRRGQ-ILWFR-GLNRI-QTQIK-1105 of human PMCA and fluorescein was attached to the C-terminal native Lys. The CaMKII peptide sequence was derived from 294-NARRK-LKGAI-LTTML-ATRN-312 of human CaMKII and fluorescein was attached to a non-native cysteine added to the N-terminus. The N-terminal His6-tagged CaM or centrin1 proteins were 2-fold diluted in a buffer composed of 20 mM Tris Cl pH 7.5, 50 mM NaCl, 1 mM CaCl_2_ and 0.005% *v*/*v* Tween 20 in a black, low volume, round bottom 384-well plate (cat. no. 4514, Corning, Amsterdam, The Netherlands). Then, 10 nM of fluorescein-labelled peptide was added to the protein dilution series. The reaction mix was incubated for 20 min at RT before anisotropy measurements.

The Sfi1 peptide was derived from 670-REVAA-RESQH-NRQLL-RGALR-RWK-692 of human Sfi1 and the fluorescein was attached to the native C-terminal Lys. Sfi1 peptide titration was performed in a buffer composed of 10 mM HEPES pH 7.4, 100 mM CaCl_2_, and 0.005% *v*/*v* Tween 20. For binding, to a 2-fold dilution series of centrin1, 100 nM of Sfi1 peptide was added, and the reaction mix was incubated for 45 min at RT before anisotropy measurements. For measuring the IC_50_ of inhibitors to centrin1, to the 3-fold dilution series of inhibitors in the assay buffer, a complex of 100 nM fluorescein labelled Sfi1 peptide and 250 nM His-centrin1 was added in 20 µL volume in a 384-well plate. The fluorescence anisotropy was measured after overnight incubation at RT.

The fluorescence anisotropy was measured on a Clariostar (BMG Labtech, Ortenberg, Germany) plate reader using the fluorescence intensity signal recorded from vertical (*I_v_*)- and horizontal (*I_h_*)-polarised light using a fluorescence polarisation module (λ_excitation_ 482 ± 8 nm and λ_emission_ 530 ± 20 nm). Fluorescence anisotropy was calculated from the measured fluorescence intensities according to $r=\frac{I_{v}-G\left(\lambda \right)I_{h}}{I_{v}+2G\left(\lambda \right)I_{h}},$ where *r* is the fluorescence anisotropy value and *I_v_* and *I_h_* are the fluorescence emission intensities detected with vertical and horizontal polarisation, respectively. The instrument specific correction factor *G*(*λ*) was set to 1 and not determined further. A quadratic equation as described [40,41] by others was defined in Prism (GraphPad, version 9.5.1, La Jolla, CA, USA) and was used to determine the *K_D_* value of the fluorescein tagged peptides to target protein.
$$y=\frac{Af+\left(Ab-Af\right)*(Lt+K_{D}+x-\sqrt{{\left(Lt+K_{D}+x\right)}^{2}-4*Lt*x}}{2Lt}$$

Here, *Af* is the anisotropy value of the free fluorescent probe, *Ab* is the anisotropy value of the fluorescent probe/protein complex, *Lt* is the total concentration of the fluorescent probe, *K_D_* is the equilibrium dissociation constant, *x* is total concentration of protein, and *y* is measured anisotropy value. *K_D_* is measured in the same unit of *x*. Note that variations in the active fraction of the home-made proteins and different methods used to determine the protein concentrations of the obtained *K_D_* values can vary from those reported.

The *IC*_50_ value of inhibitors was determined by plotting the log concentration of inhibitor against fluorescence anisotropy values and fitting the data to log inhibitor vs. response—variable slope (four parameters) equation in Prism (GraphPad). The *IC*_50_ of the inhibitor was converted into *K_d_* as described earlier using the equation [42],
$$K_{d}=\frac{{\left[I\right]}_{50}}{1+\frac{{\left[P\right]}_{50}}{\;K_{D,probe}}+\frac{{\left[E\right]}_{0}}{K_{D,probe}}}$$
where ${\left[I\right]}_{50}=IC_{50}-{\left[EI\right]}_{50}$, in which [*EI*]_50_ is the concentration of the centrin1:inhibitor complex at 50% displacement, [*I*]_50_ is the free inhibitor concentration at 50% displacement, [*P*]_50_ is the concentration of the free probe, F-Sfi1 at 50% displacement, [*E*]_0_ is concentration of free centrin1 at 0% displacement, and *K_D,probe_* is the dissociation constant of the complex of centrin1 and Sfi1.

### 2.5. Co-Immunoprecipitation Experiments

About 800,000 HEK293-ebna cells were seeded in 60 mm dishes and cultured in Dulbecco’s Modified Eagle’s Medium (DMEM) supplemented with 10% *v*/*v* Foetal Bovine Serum (FBS), 2 mM L-glutamine (cat. no. 25030-024, Gibco, Thermo Fisher Scientific), and 1% *v*/*v* penicillin/ streptomycin (cat. no. 15140122, Gibco, Thermo Fisher Scientific) overnight. The next day, cells were transiently transfected with 4 µg plasmids encoding the indicated combinations of constructs using jetPRIME (cat. no. 114-75, Polyplus, Leuven, Belgium) according to the manufacturer’s instructions. At 48 h post-transfection, the cells were lysed using 200 μL of Lysis buffer (10 mM Tris Cl pH 7.5, 150 mM NaCl, 2 mM CaCl_2_, 0.2% *v*/*v* NP40) supplemented with protease inhibitor cocktail (cat. no. A32955, Pierce, Thermo Fisher Scientific). After 30 min incubation on ice, the lysate was cleared by centrifugation for 10 min at 4 °C and 17,000× *g*. The cleared lysate was transferred to a clean tube and 15 µL sample was withdrawn (as “Input” for Western blot analysis). The lysate was diluted with 300 μL of dilution buffer (10 mM Tris Cl pH 7.5, 150 mM NaCl, 2 mM CaCl_2_) supplemented with protease inhibitor cocktail. Then 25 µL of GFP-trap Beads Slurry (ChromoTek GFP-Trap Agarose, cat. no. gta, Proteintech Europe, Manchester, UK) were added to the diluted lysate and rotated end-over-end for 1 h at 4 °C. Then, the beads were washed 3 times with Wash buffer (10 mM Tris Cl pH 7.5, 150 mM NaCl, 2 mM CaCl_2_, 0.02% *v*/*v* NP40). Bound proteins were eluted by the addition of 2 × Laemlli buffer and boiling for 10 min at 95 °C. The eluted proteins were subsequently analysed by SDS-PAGE on 10% acrylamide gels. Using the Trans-Blot Turbo Transfer system (Bio-Rad, Temse, Belgium), proteins were transferred onto a 0.2 µm nitrocellulose membrane (Bio-Rad) and incubated with a primary antibody. The following primary antibodies were used: anti-GFP (SAB4301138, Sigma-Aldrich, Overijse, Belgium, at dilution ratio 1:5000), anti-Renilla Luciferase (ab187338, Abcam, Cambridge, UK, at dilution ratio 1:3000), and anti β-actin (A5441, Sigma-Aldrich, Overijse, Belgium, at dilution ratio 1:5000). Anti-rabbit IRDye 680RD or anti-mouse IRDye 800CW secondary antibodies (LI-COR Biosciences, Bad Homburg vor der Höhe, Germany) were used to visualise the proteins on an Odyssey CLx system (LI-COR). The relative expression level of proteins was densitometrically quantified from images of membranes analysed using Image Studio software (LI-COR, version 5.2). For the quantitative analysis of the pull-down proteins, the signal of the Rluc8-tagged prey proteins was normalized with the signal from the GFP-tagged bait protein. Next, the signal intensity of the GFP2-K-RasG12V + Rluc8-CaM transfected sample was used to normalize the other samples.

### 2.6. BRET Donor Saturation Titration Assays

The detailed method of our BRET assay can be found in [9,43]. Briefly, ~200,000 HEK293-ebna cells were seeded per well of a 12-well plate (cat. no. 665180, Greiner Bio-One, Vivoorde, Belgium) and grown in 1 mL of complete DMEM. The next day, ~1 µg of BRET sensor plasmids was transfected using 2.5 µL of jetPRIME. The concentration of donor plasmid was 25 ng, and that of the acceptor plasmid had increased from 25 ng to 1000 ng for titration curves. Cells were treated with inhibitors or vehicle control (DMSO at 0.2% *v*/*v*) 24 h after transfection. Cells were collected the following day in PBS and plated in white, flat bottom 96-well plates (cat. no. 236108, Nunc, Thermo Fisher Scientific). BRET measurements were performed on a Clariostar plate reader (BMG Labtech). Three channels were read. The first channel was First, GFP2-fluorescence (λ_excitation_ 405 ± 10 nm and λ_emission_ 515 ± 10 nm), which is directly proportional to the acceptor concentration (RFU). Second, the channel was read after adding coelenterazine 400a (cat. no. C-320, GoldBio, Saint Louis, MO, USA; at 10 μM final concentration) BRET-channel (515 ± 15 nm) readings in well-mode, and then again (third) with the luminescence channel (410 ± 40 nm), and recordings were made. Signals corresponded to the BRET signal and donor (RLU) signals. The ratio of BRET signal/RLU gave the raw BRET ratio. The final BRET ratio (BRET in plots) was obtained by subtracting the raw BRET ratio from the background raw BRET ratio of cells expressing only the donor. The relative expression is calculated as the ratio of RFU/RLU and denoted as [Acceptor]/[Donor]. The BRET ratio vs [Acceptor]/[Donor] ratio data from biological repeats (typically three) were plotted together, and the data were fitted by a hyperbolic equation in Prism. The BRETtop value represents the top asymptote of the BRET ratio reached within the defined [Acceptor]/[Donor] ratio. The one phase association equation of Prism 9 (GraphPad) was used to predict the top asymptote Ymax-value, which was taken as the BRETtop. Statistical analysis between the BRETtop values was performed using the Extra sum-of-squares F test. 

### 2.7. Dose Response Analysis of Inhibitors and siRNA Knockdown in BRET Assays

For dose response analysis of inhibitors, on day one, ~200,000 HEK293-ebna cells were seeded per well of a 12-well plate (cat. No. 665180, Greiner Bio-One, Vivoorde, Belgium) and grown in complete DMEM. On day two, ~1 µg of BRET sensor plasmids were transfected at the indicated donor/acceptor plasmid ratio using jetPRIME, as mentioned in the corresponding figure legends. On day three, the medium was exchanged with fresh medium containing various doses of inhibitors. After 24 h incubation, on day four, the cells were collected in PBS, and the BRET assay was performed. The log inhibitor vs BRET ratio was plotted, and the data were fitted by a log (inhibitor) vs. response variable slope (four parameters) equation of Prism, and the IC_50_ values were calculated. 

For studying the effect of siRNA-mediated knockdown, on day one, the HEK293-ebna cells were seeded in 12-well plates in 1 mL of growth medium. On day two, cells were transfected using 3.5 µL Lipofectamine RNAiMAX (cat. no. 13778, Thermo Fisher Scientific) and Opti-MEM medium (cat. no. 31985062, Gibco, Thermo Fisher Scientific) as the vehicle with 100 nM of siRNA per well. On the next day, the medium was exchanged, and the cells were transfected with ~1 µg of BRET sensor plasmids using 3 µL jetPRIME reagent and expressed for 48 h. The transfected donor/acceptor plasmid ratio is indicated in corresponding figure legends. On day five, the BRET assay was performed as indicated above.

### 2.8. siRNA-Mediated Knockdown and Western Blotting

About 300,000 HEK293-ebna cells were seeded per well of a 6-well plate (cat. no. 657160, Cellstar, Greiner Bio-One) and grown in 2 mL of complete DMEM for 24 h. The next day, cells were transfected with 100 nM of siRNA using Lipofectamine RNAiMAX, followed by a medium exchange after 4 h. After 48 h, cells were lysed in RIPA buffer (10 mM Tris, 150 mM NaCl, 0.5 mM EDTA, 0.2% *v*/*v* NP40) supplemented with protease inhibitor cocktail (cat. no. A32955, Pierce, Thermo Fisher Scientific). The protein amount in cell lysates were quantified using Bio-Rad protein assay kit (cat. no. 5000006). Cell lysate containing 50 µg of protein per lane was resolved in Mini-PROTEAN precast 4–20% acrylamide gels. Proteins were subsequently transferred onto a nitrocellulose membrane 0.2 µm (Bio-Rad) using the Trans-Blot Turbo Transfer system (Bio-Rad) and probed with the mix of primary antibodies against the protein of interest and the loading control. The primary antibodies employed were anti-FNTA (cat. no. ab109738-1001, Abcam, at 1:1000), anti-centrin1 (cat. no. 12794-1-AP, Proteintech, Manchester, UK, at dilution ratio 1:500) and anti-GAPDH (cat. no. G8796 mouse and G9545 rabbit, Sigma-Aldrich, at 1:10,000). Anti-mouse or anti-rabbit IRDye 800CW or 680RD secondary antibodies (LI-COR) were used subsequently to develop the membrane, and the proteins were detected using an Odyssey CLx system (LI-COR).

### 2.9. Three-Dimensional Spheroid Assay

MDA-MB-231 and MCF-7 cells were seeded in 12-well plates (cat. No. 665180, Greiner Bio-One) and transfected with either 100 nM negative control siRNA or siRNA targeting *CALM1* or *CETN1* using Lipofectamine RNAiMAX. A day later, cells were harvested and plated into low-attachment, suspension cell culture 96-well plates (cat. no. 655185, Cellstar, Greiner Bio-One) for 3D spheroid suspension culture. About 1000 MDA-MB-231 or 2500 MCF-7 cell were seeded per well of the 96-well plate in 50 µL of RPMI medium (cat. no. 52400-025, Gibco, Thermo Fisher Scientific) or DMEM, respectively, containing 0.5% *v*/*v* MethoCult (cat. no. SFH4636, Stemcell technologies, Grenoble, France), 1x B27 (cat. no. 17504044, Gibco, Thermo Fisher Scientific), 25 ng/mL EGF (cat. no. E9644, Sigma-Aldrich), and 25 ng/mL FGF (cat. no. RP-8628, Thermo Fisher Scientific). Cells were incubated in a cell culture incubator for 6 days, and fresh growth medium was supplemented on the third day. After six days of incubation, the alamarBlue reagent (cat. No. DAL1025, Invitrogen, Thermo Fisher Scientific) was added to each well (10% final volume) for 4 h at 37 °C. Using a Clariostar plate reader, the fluorescence signal (λ_excitation_ 560 ± 5 nm and λ_emission_ 590 ± 5 nm) was recorded. Fluorescence signals were normalized to negative control siRNA, which was set to 100% sphere formation.

### 2.10. Confocal Microscopy

HeLa cells were seeded on glass coverslips 1.5H (cat. no. LH22.1, Carl Roth, Karlsruhe, Germany) in 6-well plates (cat. no. 657160, Cellstar, Greiner Bio-One) and grown in complete DMEM for 24 h. The next day, the cells were transiently co-transfected with pmCherry-CaM and pmGFP-K-RasG12V using jetPRIME. At 48 h after transfection, cells were fixed using 4% *v*/*v* formaldehyde (cat. no. 43368, Alfa Aesar) in PBS for 10 min at room temperature. The fixation solution was then replaced with PBS-Tween 0.05% *v*/*v* (cat. no. 9127.1, CarlRoth). After permeabilization in PBS-Triton X100 0.5% *v*/*v* (cat. no. T8787, Merck, Overijse, Belgium) for 10 min and blocking for 30 min in 2% *v*/*v* solution of BSA (A6588, Applichem, Darmstadt, Belgium) in PBS, the cells were incubated for 1 h at room temperature with primary antibody against centrin1 (rabbit polyclonal, cat no.12794-1-AP, Proteintech). After washing with PBS-Tween 0.05% *v*/*v*, the secondary antibody AlexaFluor 667 goat anti-rabbit (cat no. A21244, Life Technologies, Thermo Fisher Scientific) was applied for 1 h at room temperature. A 1 mg/mL solution of DAPI (cat. no. D1306, Thermo Fisher Scientific) in PBS for 10 min was used for DNA-staining. Using Vectashield (cat. no. H-1000, Vector Laboratories, Brussels, Belgium) coverslips were mounted onto glass slides. Images were captured on a spinning disk confocal microscope (Andor, Oxford Instruments, Belfast, UK) fitted with a Zyla 5.5 sCMOS camera (Andor, Oxford Instruments) and using a plan APO 60×/1.40 Ph3 DM oil immersion objective (Nikon, Brussels, Belgium) and NIS-Elements Imaging Software (Nikon, Version 5.42.02).

### 2.11. Data and Statistical Analysis

Prism 9 (GraphPad) was used for the preparation of plots, data, and statistical analysis. The number of independent biological repeats (n) and the type of statistical analysis used are indicated in the corresponding figure legends. A *p*-value < 0.05 is considered statistically significant, and the statistical significance levels are annotated as follows: * *p* < 0.05; ** *p* < 0.01; *** *p* < 0.001; **** *p* < 0.0001, or ns = not significant.

## 3. Results and Discussion

### 3.1. Binding Studies Support Specific Canonical Target Peptides for CaM or Centrin1

Calmodulin (CaM) and centrin proteins are highly related, both at the sequence level (Figure 1A) with 54% sequence identity between CaM and centrin1, and structurally (Figure 1B), with the most obvious difference being the N-terminal extension of centrins. While three CaM genes encode proteins with the exact same sequence, the three centrin paralogs are more divergent. Centrin1 and −2 are ~84% identical in sequence, while centrin3 differs significantly from centrin1 with only 58% sequence similarity. In several vertebrates, at least two paralog genes from each family of Ca^2+^-binding proteins are found, supporting their cell biological significance (Figure 1C).

Current evidence suggests that both CaM and centrins have distinct target protein selectivities [34]. We therefore examined whether centrin1 could also bind to classical CaM target proteins, such as the plasma membrane calcium transporting ATPase isoform 4b (PMCA) and CaM-dependent kinase II (CaMKII). PMCA removes intracellular calcium and, a 20-residue stretch mediates its regulation by CaM to which it binds with low nanomolar affinity [44]. An even higher picomolar affinity has been reported for the 19-residues of CaMKII [45]. 

We employed fluorescence polarization experiments to measure the binding of fluorescein-labelled peptides of these target proteins, F-PMCA and F-CaMKII, to His-tagged CaM and centrin1, respectively. Similar to previous observations with bovine CaM [26], we found that both peptides bound to human CaM with low nanomolar affinity (F-PMCA, K_D_ = 36 ± 5 nM; F-CaMKII, K_D_ = 6.6 ± 0.2 nM) (Figure 1D). By contrast, no binding of either peptide to human centrin1 was observed, even at 2 μM centrin1 concentration (Figure 1E). However, when testing a fluorescently labelled 18-residue long centrin1-specific target peptide, F-Sfi1, derived from the mitotic spindle regulator Sfi1, we observed a nanomolar affinity (K_D_ = 30 ± 12 nM) (Figure 1F), which was higher than the reported micromolar affinity [46]. This deviation could be partially explained by the applied methods, as in the latter case, isothermal titration calorimetry was used.

Overall, these results suggest that the sequence divergence between CaM and centrin1 is sufficient to define specific binding to their classical targets that contain a distinct peptide recognition sequence.

### 3.2. Cellular BRET Data Suggest That the K-Ras G-Domain Participates in Complexes with Either CaM or Centrin1

Given the high sequence similarity between CaM and centrin1 (Figure 1A), we investigated whether farnesylated K-Ras could bind to centrin1 as a non-canonical target. Centrin1 was chosen due to its expression in ciliated cells, notably stem cells [32,47]. We therefore established a cellular Bioluminescence Resonance Energy Transfer (BRET)-assay to test binding of wild-type K-Ras or oncogenic K-RasG12V to centrin1 as compared to CaM.

We genetically fused the donor emission enabling Renilla Luciferase-derivative Rluc8 to the N-terminus of the K-Ras protein and the acceptor GFP2 to the N-terminus of centrin1 or CaM. If donor- and acceptor-tagged proteins interact, the BRET signal increases with increasing acceptor-to-donor ratio and may reach a saturation value. Commonly, the BRETmax value describes an absolute saturation value [48], which is typically not reached in most BRET titration experiments, and is therefore associated with significant extrapolation.

We here introduce the BRETtop value that characterizes the highest BRET value reached within a defined acceptor-to-donor ratio titration range. By keeping the titration range constant, we can compare different BRETtop values with each other. As with FRET, the BRET-values depend on the distance between the luminophores in the complex of interacting proteins. Therefore, only if the binding modes, i.e., the structure of the complexes are comparable, such as can be reasonably assumed for point mutants or paralogs of a protein, higher BRETtop values indicate a higher interaction probability and strength of examined BRET-pairs in cells. Like BRETmax, BRETtop would then correlate with the relative number of binding-sites and the relative affinities.

**Figure 1 cancers-15-03087-f001:**
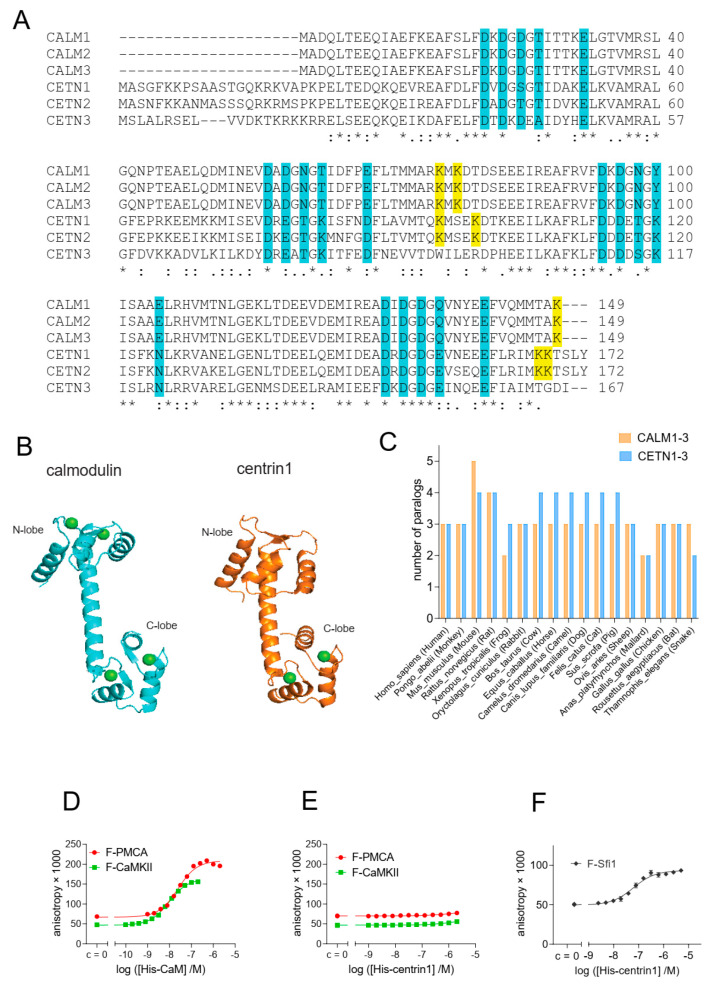
Despite its high similarity to CaM, centrin1 does not recognize CaM-target peptides. (**A**) Multiple sequence alignment of human CaM (*CALM1-3*) and centrin (*CETN1-3*) protein paralogs, designated by the encoding gene names. The Ca^2+^-binding residues are highlighted in cyan. Lysines 75, 77, and 148 of CaM, which become covalently modified by CaM inhibitor ophiobolin A, are highlighted in yellow. The same highlight was used for lysine residues at similar positions in centrin1 and centrin2, while no such lysine residues could be identified for centrin3. Note that the CaM protein numbering starts at Ala, as the N-terminal, and native Met is removed in most organisms [49]. (**B**) Structures of human CaM (PDB ID 1CLL) and human centrin1 (PDB ID 2GGM). Calcium ions are marked as green spheres. Structures were generated using the PyMOL Molecular Graphics System, Version 2.4.0, Schrödinger, LLC. (**C**) Analysis of the number of paralog coding genes of *CALM1-3* and *CETN1-3* in different species. Data were curated from the NCBI protein database. (**D**–**F**) Binding of 10 nM fluorescein-labelled F-CaMKII and F-PMCA (**D**,**E**) or 100 nM F-Sfi1 (**F**) peptides to His-tagged human CaM or centrin1 was detected using fluorescence anisotropy measurements. * *p* < 0.05; ** *p* < 0.01; *** *p* < 0.001; **** *p* < 0.0001.

We previously observed a higher interaction BRET-signal of oncogenic K-Ras as compared to its wild-type (wt) counterpart with CaM [9]. In line with these data, both CaM and centrin1 were significantly more co-immunoprecipitated with oncogenic GFP2-tagged K-RasG12V than with wt K-Ras (Figure 2A,B; Appendix A).

Consistent with our previous BRET-data, we also found that K-RasG12V had a significantly higher BRETtop with CaM than wt K-Ras (Figure 2C). Similarly, the BRETtop of K-RasG12V with centrin1 was significantly higher than that of wt K-Ras with centrin1 (Figure 2D). As expected, a control BRET-pair showed significantly lower BRET-values than the weakest BRET-interaction pair studied (Appendix A). The higher BRET of K-RasG12V with the Ca^2+^-binding proteins was surprising given the afore-mentioned in vitro binding data [15,25]. Ras binding to some effectors can be reduced by the D38A-mutation, which abolishes major contacts preserved in several effector complexes [50,51]. Addition of the D38A mutation reduced the BRET to the level of wt K-Ras for both CaM and centrin1 (Figure 2C,D). This may suggest a dependence on some effectors or other effector lobe binders; however, more comprehensive studies are required to demonstrate this.

### 3.3. CaM Inhibitors Bind to Centrin

Given that centrin1 possesses lysines on positions 96, 100, 167, and 168 that are homologous to those targeted by covalent CaM inhibitors (Figure 1A), we tested whether covalent CaM inhibitors ophiobolin A and calmirasone1 or the potent non-covalent CaM inhibitor calmidazolium would disrupt binding of K-Ras to centrin1 in cells. Indeed, treatment with any of these CaM inhibitors lowered the BRETtop of K-RasG12V/centrin1 (Figure 3A). The inhibition of this interaction occurred at IC_50_ (calmidazolium) = 10.44 ± 0.05 µM and IC_50_ (calmirasone1) = 41.6 ± 0.3 µM (Figure 3B), the latter of which was comparable to what was previously observed with CaM [9]. For centrin1, fluorescence anisotropy data revealed that CaM inhibitors can displace fluorescently labelled Sfi1 from it, indicating their direct binding to centrin1 (Figure 3C, Table 1). 

The sensitivity of the K-Ras/centrin1 interaction to CaM inhibitors suggests conserved inhibitor binding sites and a similar mode of interaction between K-Ras and the Ca^2+^-binding proteins. Importantly, treatment with several CaM inhibitors may therefore also affect centrin1 biology, making it potentially difficult to interpret inhibitor-dependent phenotypic observations.

### 3.4. Inhibition of Prenylation Does Not Disrupt the BRET-Interaction of K-Ras with CaM or Centrin1 in Cells

Agamasu et al. have previously reported that the K-Ras-derived farnesylated and carboxymethylated KSKTKC-peptide is sufficient to bind to CaM in vitro [25]. To test whether non-prenylated K-Ras can still bind to the Ca^2+^-binding proteins in cells, we tested the effects of the prenylation inhibitor mevastatin in our BRET-assays. Statins such as mevastatin inhibit the HMG-CoA pathway, and thus, provision of prenylpyrophosphate substrates for protein prenylation [52]. We therefore expected that treatment of cells with high concentrations of mevastatin would abrogate farnesyl-mediated K-Ras/CaM interaction. Surprisingly, mevastatin treatment did not significantly affect the BRET-levels of K-Ras with either CaM or centrin1 (Figure 4A,B). The higher BRETtop of K-Ras with CaM (Figure 4A) than with centrin1 (Figure 4B) may relate to the fact that only one Ca^2+^-binding lobe is found in centrins [32], which may allow for the binding of only one K-Ras per centrin1 protein.

We next examined whether the C-terminal membrane targeting sequence of K-Ras alone (residues 166–188), CTK, was sufficient to mediate binding to CaM, as suggested by in vitro data, and whether the same would apply for binding to centrin1. In agreement with in vitro data, the BRET between CTK and CaM indicated binding; it had a lower BRETtop than full length K-Ras (Figure 4C), but the BRET-values were still above background (Appendix A). The CTK interaction with centrin1 was comparable to that with CaM (Figure 4C,D). As with full-length K-Ras, mevastatin treatment did not decrease the BRET of CTK with either of the Ca^2+^-binding proteins (Figure 4C,D).

This mevastatin insensitivity was overall unexpected, given the strong contribution of the farnesyl-moiety to CaM-binding in vitro [15,24,25], but it was in line with data showing binding of non-farnesylated K-RasG12V to CaM in vitro [53]. 

Taken together with the activation-state dependent complexation of K-Ras with CaM or centrin1, this may suggest that these proteins exist in cellular complexes that are largely prenylation independent yet involve the C-terminal poly-lysine stretch of K-Ras and depend on the activation state of K-Ras. Alternatively, similarly sized, distinct pools of K-Ras in complex with the Ca^2+^-binding proteins exist, and they require a subset of the aforementioned features.

### 3.5. Membrane Targeting and Anchorage of K-Ras Depends More on CaM Than on Centrin1

Prenyl-binding chaperone proteins can effectively facilitate diffusion of their target proteins in cells, as they shield the hydrophobic prenyl-moiety and thus allow for a longer residence in the aqueous cytoplasm [4]. Others suggested that CaM can extract and solubilize K-Ras and act as a trafficking chaperone [15]. 

We previously showed that inhibition of CaM selectively reduces K-RasG12V- as compared to H-RasG12V-BRET signals that originate from nanoclustering of active Ras on the plasma membrane [9]. This nanoclustering-dependent BRET-signal is sensitive to disruption, not only of Ras nanoclustering, but of any process upstream that interferes with functional membrane anchorage, such as disrupted trafficking or inhibition of the Ras lipid modification [54]. Similar to CaM inhibition, knockdown of another trafficking chaperone, PDE6D, which also binds to prenylated proteins and facilitates K-Ras localization at the plasma membrane, reduces K-Ras membrane anchorage associated FRET [55]. 

In line with the mevastatin data (Figure 4), binding of K-Ras to CaM or centrin1 was essentially insensitive to inhibition of prenylation by the knockdown of the shared α-subunit of farnesyl- and geranylgeranyl-transferases (*FNTA*) (Figure 5A–C). However, the same treatment significantly abrogated the membrane anchorage-BRET signal of both K-RasG12V or H-RasG12V (Figure 5E,F), consistent with the significance of prenylation for Ras membrane anchorage [56]. 

As observed previously, knockdown of CaM selectively reduced the membrane anchorage-BRET signal of K-RasG12V (Figure 5E) but not H-RasG12V (Figure 5F). By contrast, knockdown of centrin1 decreased the BRET-signal K-Ras-selectively and to a significantly lesser extent than knockdown of CaM (Figure 5D–F). Immunoblotting confirmed the significant knockdown of *FNTA* and centrin1 (*CETN1)* expression in HEK293-ebna cells (Figure 5C,D; Appendix A), while that of CaM (*CALM1)* was previously validated by us using RT-qPCR [9]. 

These data suggest that CaM is more important to facilitate membrane trafficking of K-Ras in cells than centrin1.

### 3.6. Centrin1 Co-Distributes with CaM during the Cell Cycle

We previously observed that CaM inhibitors decrease stemness properties of KRAS-mutant cancer cell lines [8,9]. The clonogenic growth of cancer cell spheroids is employed as a surrogate measure for cancer cell stemness [57]. We therefore tested the effect of the knockdown of CaM (*CALM1*) and centrin1 (*CETN1*) on MDA-MB-231 and MCF-7 derived spheroids (Figure 6A,B). Both *CALM1* and *CETN1* (Appendix A) knockdown decreased the formation of spheroids derived from these cell lines. However, the effect was more pronounced in the KRAS-mutant MDA-MB-231 cell line (Figure 6A). Moreover, the knockdown of *CALM1* decreased spheroid growth significantly more in this cell line, which correlated with the overall stronger effect of this knockdown treatment on K-RasG12V membrane anchorage BRET (Figure 5E).

Only limited conclusions in regard to cancer cell stemness can be derived from these experiments, which are essentially assaying the ability of cancer cells to evade anoikis and which bear some similarity to culture conditions employed for stem/progenitor cells.

The fate of stem and progenitor cells is decided during the cell cycle, which can proceed to symmetric or asymmetric cell divisions [58]. Oncogenes are suggested to shift the mode of cell division that has more symmetric divisions and produce more stem cells [59]. Stemness can be mediated by centriolar organelles, such as the centrosomes, specifically the mother centrosome [60]. Interestingly, mCherry-tagged CaM localises during different cell cycle phases to the centrosomes and the midbody in HeLa cells (Figure 6C), as observed previously by others [19,27]. The same is essentially seen for endogenous centrin1, which also localises to these structures (Figure 6C). However, during interphase, CaM has a more pronounced cyto/nucleoplasmic distribution, while centrin1 discretely localises to the centrosomes.

## 4. Conclusions

Our data show that K-Ras does not only interact with CaM, but also with the highly related protein centrin1. While both Ca^2+^-binding proteins distribute to similar mitotic structures, notably the centrosomes, CaM appears to have a stronger impact on K-Ras functional membrane organisation at the plasma membrane. Centrin1 may instead function to localize K-Ras to certain structures, such as the centrosomes, while it appears to have only a minor role in K-Ras trafficking. These distinct functions of CaM and centrin proteins are difficult to tell apart using pharmacological inhibitors against CaM, which we found affect binding of K-Ras to centrin1 as well.

While previous in vitro data demonstrated that the farnesylated C-terminus of K-Ras was sufficient for binding to CaM, our data here suggest that farnesylation is essentially dispensable for most of the interaction of K-Ras with either CaM or centrin1 in cells. Given that oncogenic K-Ras engages more with either of these proteins, we propose that most of the K-Ras/CaM and K-Ras/centrin1 pairs are found in complexes that can recognize the activation state of K-Ras. This recognition is typically afforded by effectors, hence it is plausible to assume that most of the K-Ras binding to these Ca^2+^-binding proteins happens in higher order complexes that contain effectors. Others have previously proposed PI3Kα-containing complexes with K-Ras and CaM [61]. Our data encourage further investigation of these potential complexes and their function inside cells.

## Figures and Tables

**Figure 2 cancers-15-03087-f002:**
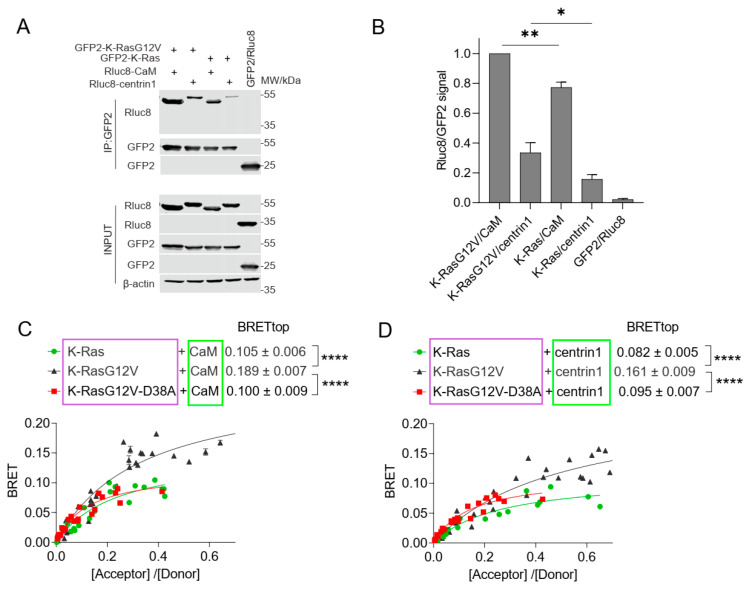
The interaction of CaM or centrin is increased with oncogenic K-Ras. (**A**) Co-immunoprecipitation of Rluc8-CaM or Rluc8-centrin1 with GFP2-K-RasG12V or GFP2-K-Ras wt. Pull-down was performed using lysates of HEK293-ebna cells transfected with combinations of GFP2-K-RasG12V/Rluc8-CaM, GFP2-K-RasG12V/Rluc8-centrin1, GFP2-K-Ras/Rluc8-CaM, GFP2-K-Ras/Rluc8-centrin1 and GFP2/Rluc8 and expressed for 48 h. The GFP2-tagged protein was bound using GFP-trap beads, and the samples were analysed using anti-Rluc8 and anti-GFP antibodies. See Appendix A for the original images of Western blots. (**B**) Immunoprecipitated Rluc8-tagged protein signals were normalized to GFP-tagged protein signals. The signal intensity of the GFP2-K-RasG12V/Rluc8-CaM transfected sample was set to 1 in each experiment and was used to normalize the other samples. The plot shows mean ± SEM and the statistical analysis was performed using one-way ANOVA test. (**C**,**D**) Interaction of Rluc8-K-Ras wt, Rluc8-K-RasG12V, and Rluc8-K-RasG12V-D38A with GFP2-CaM (**C**) or GFP2-centrin1 (**D**). All samples were treated with 0.2% *v*/*v* DMSO for 24 h, n = 3. Statistics of BRETtop values were analysed using the F-test. BRET donor protein is boxed purple, acceptor protein is boxed green. * *p* < 0.05; ** *p* < 0.01; **** *p* < 0.0001.

**Figure 3 cancers-15-03087-f003:**
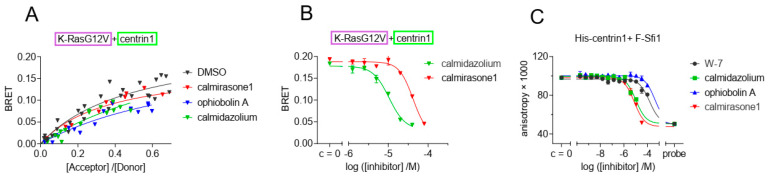
The interaction of K-Ras with centrin1 is modulated by direct binding of CaM inhibitors to centrin1. (**A**) HEK293-ebna cells were transfected with Rluc8-K-RasG12V/GFP2-centrin1 BRET sensor plasmids for 24 h followed by treatment with ophiobolin A (2.5 µM), calmidazolium (10 µM), calmirsone1 (20 µM) or equal volume of DMSO (0.2% *v*/*v*) for another 24 h, n = 3. (**B**) HEK293-ebna cells were transfected with BRET sensor plasmids Rluc8-K-RasG12V/ GFP2-centrin1 at a ratio of 1/19, respectively, for 24 h followed by a 24 h treatment with 2-fold dilution series of calmidazolium or calmirasone1 ranging from 80 µM to 0.1 µM. Data represent mean ± SEM, n = 2. BRET donor protein is boxed purple, acceptor protein is boxed green. (**C**) Displacement of fluorescent F-Sfi1 from centrin1 by CaM inhibitors. The inhibitors were 3-fold diluted in assay buffer, followed by addition of the complex of 100 nM F-Sfi1 and 250 nM His-centrin1. The fluorescence anisotropy was measured after overnight incubation at RT.

**Figure 4 cancers-15-03087-f004:**
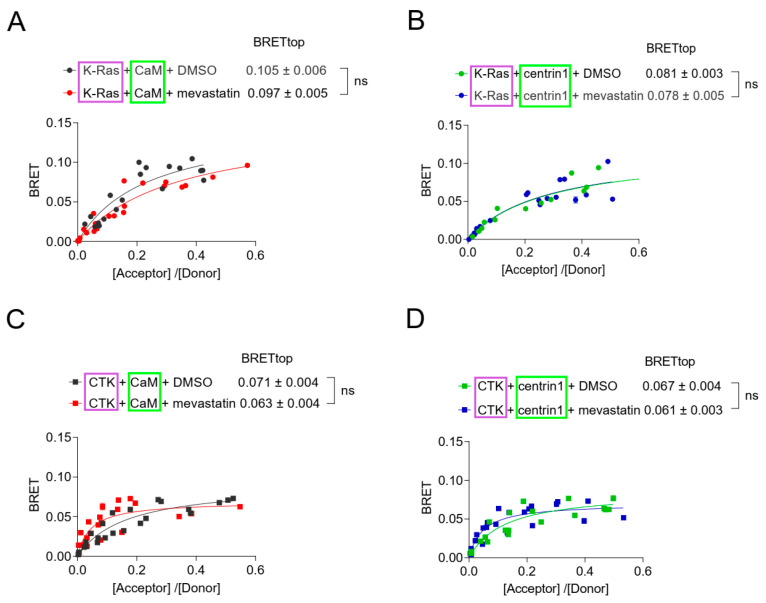
Inhibition of prenylation by mevastatin does not disrupt BRET between K-Ras and CaM or centrin1. (**A**–**D**) BRET-sensors Rluc8-K-Ras/GFP2-CaM (**A**), Rluc8-K-Ras/GFP2-centrin1 (**B**), Rluc8-CTK/GFP2-CaM (**C**) and Rluc8-CTK/GFP2-centrin1 (**D**) were transfected into HEK293-ebna cells, and cells were treated with 10 µM mevastatin or the vehicle control, DMSO 0.2% *v*/*v* for 24 h, n = 3. Statistics of BRETtop values was analysed using the F-test. BRET donor protein is boxed purple, acceptor protein is boxed green. ns = not significant.

**Figure 5 cancers-15-03087-f005:**
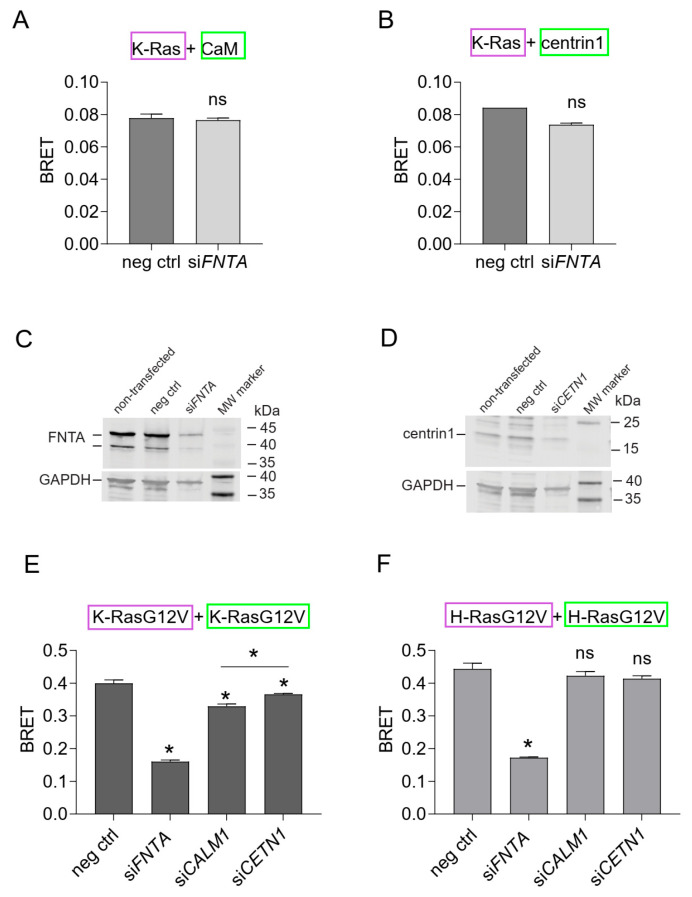
K-Ras membrane anchorage is selectively affected by CaM-, but less so by centrin1-depletion. (**A**,**B**) Rluc8-K-Ras was transfected with GFP2-CaM (**A**) or GFP2-centrin1 (**B**) plasmids at a donor/acceptor plasmid ratio of 1/5 into HEK293-ebna cells. BRET donor protein is boxed purple, acceptor protein is boxed green. Data represent mean ± SEM, n = 2 to 4. Statistical significance between negative control siRNA and sample siRNA was analysed using Mann–Whitney test. (**C**,**D**) HEK293-ebna cells were transfected with 100 nM of negative control siRNA or si*FNTA* or si*CETN1* for 48 h and cell lysates were immunoblotted as indicated. See Appendix A for the original images of Western blots. (**E**,**F**) HEK293-ebna cells were transfected with 100 nM siRNA for 24 h, followed by BRET sensor transfection. Rluc8-/GFP2-tagged K-RasG12V (**E**) or H-RasG12V (**F**) nanoclustering-BRET sensor plasmids were transfected at a donor/acceptor plasmid ratio of 1/15. BRET donor protein is boxed purple, acceptor protein is boxed green. Data represent mean ± SEM, n = 4. Statistical significance between negative control siRNA and sample siRNA was analysed using Mann–Whitney test. * *p* < 0.05; ns = not significant.

**Figure 6 cancers-15-03087-f006:**
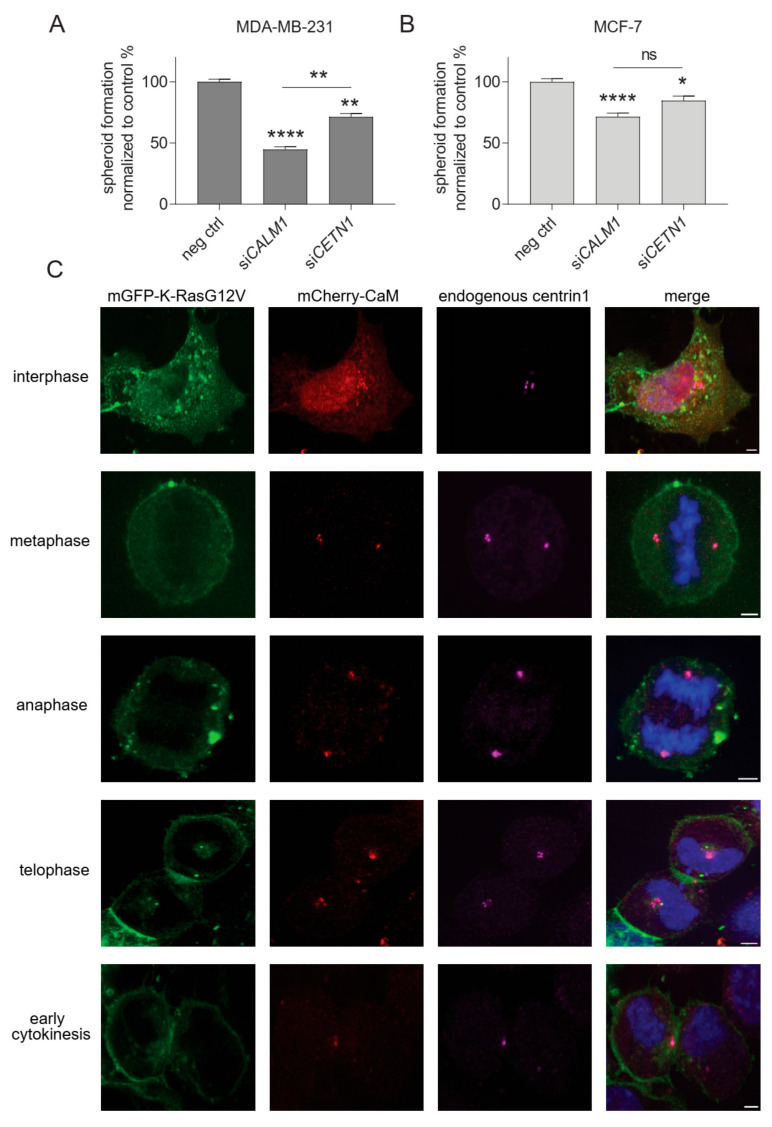
CaM and centrin1 distribute to the centrosomes, and their loss more potently affects spheroid formation of *KRAS* mutant MDA-MB-231 cells. (**A**,**B**) Effect of knockdown of CaM (*CALM1*) and centrin1 (*CETN1*) on spheroids derived from MDA-MB-231 (**A**) and MCF-7 (**B**) cells. The knockdown efficiency was compared to spheroids grown from negative control siRNA transfected cells. Data represent mean ± SEM of four biological repeats. Statistical analysis was performed using Mann–Whitney test. (**C**) Representative images of HeLa cells that were co-transfected with mGFP-K-RasG12V (green) and mCherry-CaM (red). Endogenous centrin1 was immunostained (purple), and DNA was stained using DAPI (blue). Cell-cycle stages are indicated on the left. Scale bar is 5 µm. * *p* < 0.05; ** *p* < 0.01; **** *p* < 0.0001; ns = not significant.

**Table 1 cancers-15-03087-t001:** Comparison of K_d_ values of CaM inhibitors with centrin1 and CaM determined by fluorescence anisotropy measurements. The competition assay derived K_d_ values of inhibitors to CaM were previously reported by us, using F-PMCA peptide as probe.

Inhibitor	Centrin1	CaM
	Mean K_d_ (Repeat Values)	K_d_ (References)
calmidazolium	1.6 (1.4; 1.8) µM	13.5 nM [26]
W-7	18.2 (17.8; 18.5) µM	1.47 µM [26]
ophiobolin A	49 (58; 39) µM	3.5 µM [9]
calmirasone1	0.9 (1.0; 0.8) µM	0.87 µM [9]

## Data Availability

Any data that support the findings of this study are included within the article and Appendix A.

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
