# Peer review of "K-Ras Binds Calmodulin-Related Centrin1 with Potential Implications for K-Ras Driven Cancer Cell Stemness"

_cancers, 2023, doi:10.3390/cancers15123087_

Round 1
Reviewer 1 Report
Given the importance of K-Ras in cancer biology, the manuscript by Manoharan et al. describing K-Ras binding to calmodulin and centrin1 is interesting and significant. I think this is worth publishing. However, I also have a number of concerns that I would like to be addressed. These are summarized below.
I disagree with the interpretation of BRETtop values as indicators of interaction probability and strength, as stated on line 394. This is only correct when the compared peptides have the same binding modes. If the binding modes are different, higher BRETtop values could suggest a shorter distance between the donor and acceptor and not necessarily a difference in interaction probability and strength.
There is a similar concern with the statement on lines 420-422 “Consistent with a binding contribution mediated via some effector interaction, addition of the D38A effector-site mutation reduced the BRET to the 421 level of wt K-Ras for both CaM and centrin1 (Figure 2C,D).” The mutation of D38 could have altered the structure of K-Ras and changed the peptide binding modes, leading to longer distances between the BRET donor and the BRET acceptor.
In section 3.3, is there evidence that the structural integrity of centrin1 is maintained after its covalent modification with calmirasone1 and ophiobolin A?
Regarding the statement on lines 497 and 498 “This mevastatin insensitivity was overall unexpected, given the strong contribution of the farnesyl-moiety to CaM-binding in vitro (15, 24, 25)”, please consider the publication by Abdelkarim et al., 2021 (PMID: 34311289) that reports a lower Kd value for calmodulin binding to non-modified G12V K-Ras than those reported for calmodulin binding to prenylated K-Ras in references 15, 24, and 25.
Author Response
Reviewer #1:
Given the importance of K-Ras in cancer biology, the manuscript by Manoharan et al. describing K-Ras binding to calmodulin and centrin1 is interesting and significant. I think this is worth publishing. However, I also have a number of concerns that I would like to be addressed. These are summarized below.
1- I disagree with the interpretation of BRETtop values as indicators of interaction probability and strength, as stated on line 394. This is only correct when the compared peptides have the same binding modes. If the binding modes are different, higher BRETtop values could suggest a shorter distance between the donor and acceptor and not necessarily a difference in interaction probability and strength.
We thank reviewer for this insightful comment. The reviewer is absolutely correct that this statement needs to be modified. We are now writing in L. 403ff:
As with FRET, the BRET-values depend on the distance between the luminophores in the complex of interacting proteins. Therefore, only if the binding modes i.e., the structure of the complexes are comparable, such as can be reasonably assumed for point mutants or paralogs of a protein, higher BRETtop values indicate a higher interaction probability and strength of examined BRET-pairs in cells. Like BRETmax, BRETtop would then correlate with the relative number of binding-sites and the relative affinities.
2- There is a similar concern with the statement on lines 420- 422 “Consistent with a binding contribution mediated via some effector interaction, addition of the D38A effector-site mutation reduced the BRET to the 421 level of wt K-Ras for both CaM and centrin1 (Figure 2C,D).” The mutation of D38 could have altered the structure of K-Ras and changed the peptide binding modes, leading to longer distances between the BRET donor and the BRET acceptor.
Based on the structure of Ras for instance with the Ras binding domain (RBD) of the effector RGS, Asp38 is central for contacts with effectors (Vetter IR 1999 PMID 10371160). Hence the D38A mutation of Ras is well and established and common mutation to reduce the interaction of several effectors and hence study effector dependence (Herrmann C 1996 PMID: 8636102). None of this work suggests that the mutation would alter the structure of the K-Ras/ effector complex in the sense that it would affect the orientation of effector binding.
To clarify this, we therefore now add the sentence in L. 434:
Ras binding to its effectors is commonly drastically reduced by the D38A-mutation, which abolishes major contacts preserved in several effector complexes [50,51]. Consistent with a binding contribution mediated via the effector binding region of K-Ras, addition of the D38A mutation reduced the BRET to the level of wt K-Ras for both CaM and centrin1 (Figure 2C,D).
3- In section 3.3, is there evidence that the structural integrity of centrin1 is maintained after its covalent modification with calmirasone1 and ophiobolin A?
Because these are expected to be covalent modifications, the structure will be modified in a distinct manner, not by completely unfolding the protein. We previously provided evidence that this involves lysine residues as highlighted in Fig. 1A (Okutachi S 2021, Ref. 9). Other CaM inhibitors are changing the conformation of CaM quite substantially, as we introduce in L. 86.
We find that the binding of a centrin1 with a peptide derived from a known centrin1 interactor, Sfi1, is disrupted by these inhibitors in vitro (Fig. 3C), in a very similar manner as CaM is (Okutachi S 2021, Ref. 9). At the same time these inhibitors also affect the interaction of K-Ras and centrin1 in a dose dependent manner in cells (Fig. 3B). It is most plausible to assume that the inhibitors affect the active fraction of centrin1 in both assays in an analogous fashion as they affect CaM.
4- Regarding the statement on lines 497 and 498 “This mevastatin insensitivity was overall unexpected, given the strong contribution of the farnesyl-moiety to CaM-binding in vitro (15, 24, 25)”, please consider the publication by Abdelkarim et al., 2021 (PMID: 34311289) that reports a lower Kd value for calmodulin binding to non-modified G12V K-Ras than those reported for calmodulin binding to prenylated K-Ras in references 15, 24, and 25.
We thank the reviewer for pointing us to this article, which is now cited as Ref. 53. We now write in L. 513ff.
This mevastatin insensitivity was overall unexpected, given the strong contribution of the farnesyl-moiety to CaM-binding in vitro [15,24,25], but it was in line with data showing binding of non-farnesylated K-RasG12V to CaM in vitro[53].
Reviewer 2 Report
The authors studied the binding of centrin1 to the K-Ras protein and compared it with the calmodulin-related binding capacity.
Few comments:
Why did the authors study centrin-1, and not centrin 2 or 3? Centrin-1 would not show co-localization with mutant K-ras, based on its tissue expression pattern, correct?
3.1 – title, lines 346-47. Sentence is unclear to this reviewer.
Line 357, one could add to sentence …if classical CaM target proteins, namely PCMA, CaMKII, and Sfi1, ….
Line 364: The authors describe an experimental approach and refer to a published paper (26). As written, it gives the impression that this research has already been published.
Line 565: same problem. Authors describe Figure 6C but also refer to two references. Did they already publish the data? If so, this should be stated?
Table 1: authors report mean +/- SEM for duplicates. One cannot report SEM for duplicates; it needs at least three experiments. Please report mean and the two measurements in the table.
Line 550: For consistency, the reviewer recommends writing CaM (CALM1) and centrin1 (CETN1), instead of only providing the gene symbols.
Author Response
Reviewer #2:
The authors studied the binding of centrin1 to the K-Ras protein and compared it with the calmodulin-related binding capacity.
Few comments:
1- Why did the authors study centrin-1, and not centrin 2 or 3? Centrin-1 would not show co-localization with mutant K- ras, based on its tissue expression pattern, correct?
K-Ras is ubiquitously expressed, hence also in tissues where centrin-1 is expressed, such as the neurons and ciliated cells. Given that the latter are typically found in stem and progenitor cells, these are of particular interest to us in this context. Furthermore, given the high sequence similarity between centrin1 and centrin2, it is plausible to assume our results also apply to more widely expressed centrin2.
We therefore now explain in L. 390:
Centrin1 was chosen, given its expression in ciliated cells, notably stem cells [32,47].
2- 3.1 – title, lines 346-47. Sentence is unclear to this reviewer.
We apologize for the complicated title of 3.1 and have now simplified it to:
3.1. Binding studies support specific canonical target peptides for CaM and centrin1
3- Line 357, one could add to sentence ...if classical CaM target proteins, namely PCMA, CaMKII, and Sfi1, ....
We thank the reviewer for this suggestion and write now in L. 367:
We therefore examined if centrin1 could also bind to classical CaM target proteins, such as the plasma membrane calcium transporting ATPase isoform 4b (PMCA) and CaM-dependent kinase II (CaMKII).
Please note that Sfi1 is a target of centrin1.
4- Line 364: The authors describe an experimental approach and refer to a published paper (26). As written, it gives the impression that this research has already been published.
We have removed the confusing citation, as in the subsequent sentence we correctly refer to results from that publication.
5- Line 565: same problem. Authors describe Figure 6C but also refer to two references. Did they already publish the data? If so, this should be stated?
Thank you for pointing this out. We have now added in L. 583:
... (Figure 6C), as observed previously by others [19,27].
6- Table 1: authors report mean +/- SEM for duplicates. One cannot report SEM for duplicates; it needs at least three experiments. Please report mean and the two measurements in the table.
We thank the reviewer for pointing this out. We now report in table 1 the mean Kd and the individual repeat values.
7- Line 550: For consistency, the reviewer recommends writing CaM (CALM1) and centrin1 (CETN1), instead of only providing the gene symbols.
We thank the reviewer for this good suggestion. We have implemented this at the beginning of a paragraph, in legends and in method sections, where these abbreviations are first used and as appropriate.
Round 2
Reviewer 1 Report
Thank you for addressing my comments. I recommend to use more cautious language (more general) describing the effects of the D38A mutation that are likely more nuanced than the authors think. Please look at this paper on bioRxrv.org "Structure-based prediction of Ras-effector binding affinities and design of branchegetic interface mutations" by Philipp Junk and Christina Kiel.
Author Response
Point by point response to revision on manuscript No.: cancers-2407877 round 2
K-Ras binds calmodulin-related centrin1 with potential implications for K-Ras driven cancer cell stemness
Reviewer #2:
Thank you for addressing my comments. I recommend to use more cautious language (more general) describing the effects of the D38A mutation that are likely more nuanced than the authors think. Please look at this paper on bioRxrv.org "Structure-based prediction of Ras-effector binding affinities and design of branchegetic interface mutations" by Philipp Junk and Christina Kiel.
We thank the reviewer for bringing this preprint to our attention, certainly an interesting study. We agree that D38A-sensitivity does not unequivocally prove involvement of all effectors and other effector lobe dependent interactions could probably be impacted as well.
We therefore try to apply more cautious language in the sentence in L. 433:
Ras binding to some effectors can be reduced by the D38A-mutation, which abolishes major contacts preserved in several effector complexes [50,51]. Addition of the D38A mutation reduced the BRET to the level of wt K-Ras for both CaM and centrin1 (Figure 2C,D). This may suggest a dependence on some effectors or other effector lobe binders; however, more comprehensive studies are required to demonstrate this.